# (±)-Cryptamides A–D, Four Pairs of Novel Dopamine Enantiomer Trimers from the Periostracum Cicadae

**DOI:** 10.3390/molecules27196707

**Published:** 2022-10-09

**Authors:** Junjian Luo, Wenjun Wei, Pan Wang, Tao Guo, Suiqing Chen, Liping Zhang, Shuying Feng

**Affiliations:** 1School of Pharmacy, Henan University of Chinese Medicine, Zhengzhou 450046, China; 2Academy of Chinese Medical Sciences, Henan University of Chinese Medicine, Zhengzhou 450046, China; 3Medical College, Henan University of Chinese Medicine, Zhengzhou 450046, China

**Keywords:** Periostracum cicadae, dopamine trimers, enantiomer, structure identification, anti-inflammatory

## Abstract

Four pairs of novel dopamine enantiomer trimers, (±)-cryptamides A–D (**1**–**4**), and 10 pairs of previously described dopamine enantiomer dimers (**5**–**14**) were isolated from the Periostracum cicadae, the cast-off shell of the insect *Cryptotympana pustulata*. Aside from being pairs of enantiomers, the eight trimers were also elucidated to be regioisomers, most likely resulting from their mechanism of formation, [4 + 2] cycloaddition. The discovery of dopamine trimers is rarely reported when it comes to natural products derived from insects.

## 1. Introduction

Periostracum cicadae (PC), the cast-off shell of the insect *Cryptotympana pustulata* Fabricius, is a traditional Chinese medicine, called “chantui”, and is mainly produced in the Shandong, Henan, Hebei, Hubei, Jiangsu, and Sichuan provinces [1]. PC has been used in traditional Chinese medicine for heat-clearing medicine within clinics [2]. Previously described pharmacological studies demonstrate anti-convulsion, anti-inflammatory, anti-tumor, antitussive, expectorant, anti-asthmatic, immune-suppressive, and anti-allergic properties of PC [3,4,5,6]. Previously reported chemical investigations have shown that the most common molecular components isolated from PC are *N*-acetyldopamine (NADA) derivatives [2,5,6,7,8,9,10]. At this point, it should be mentioned that most studies report findings of NADA dimers [2,5,6,8,9,10,11] and tetramers [12], while the discovery of trimers is unusual.

Hereby, we report the isolation and structural elucidation of four novel enantiomeric dopamine trimer pairs (**1**–**4**) and 10 enantiomeric dopamine dimer pairs (Figure 1), the latter of which are already described in the literature [2,10,11,13,14,15]. A detailed comparison revealed that the novel four pairs of compounds were also regioisomers. Moreover, the anti-inflammatory effects of new compounds toward a lipopolysaccharide (LPS)-induced RAW264.7 macrophage model were evaluated. Among them, (±)-cryptamide B, (±)-cryptamide C, and (−)-cryptamide D can effectively reduce the levels of nitric oxide.

## 2. Results

### 2.1. Structure Elucidation

The methanol extract of PC was eluted with MeOH-H_2_O to provide five fractions (A–E). Fractions C and D were separated using various chromatographic techniques, producing four pairs of novel dopamine enantiomer trimers (**1**–**4**) and 10 pairs of previously reported dopamine enantiomer dimers (**5**–**14**). These known compounds were identified as (±)-cicadamide E (**5**) [2], (±)-cicadamide B (**6**) [2], (±)-parvamide B (**7**) [11], (±)-(2*R*,3*S*)-2-(3′,4′-dihydroxyphenyl)-3-acetylamino-7-(*N*-acetyl-2″-aminoethyl)-1,4-benzodioxane (**8**) [10], (±)-(2*R*,3*S*)-2-(3′,4′-dihydroxyphenyl)-3-acetylamino-6-(*N*-acetyl-2″-aminoethyl)-1, 4-benzodioxane (**9**) [10], (±)-(2*R*,3*S*)-2-(3′,4′-dihydroxyphenyl)-3-acetylamino-6-(*N*-acetyl-2″-aminethylene)-1,4-benzodioxane (**10**) [10], (±)-(2*R*,3*S*)-2-(3′,4′-dihydroxyphenyl)-3-acetylamino-7-(*N*-acetyl-2″-aminethylene)-1,4-benzodioxane (**11**) [10], (±)-plancyamide B (**12**) [13], (±)-molossusamide C (**13**) [14], and (±)-aspongopusamide A (**14**) [15], based on the comparison of their spectroscopic data with the values in the literature.

Compound **1** was obtained as a white amorphous powder. Its molecular formula was determined to be C_30_H_31_N_3_O_9_, as deduced from HR-EIS-MS data (*m/z* 578.2139 [M + H]^+^ and calculated for C_30_H_32_N_3_O_9_, 578.2132). The ^1^H-NMR data (Table 1) of **1** exhibited nine proton signals of benzene rings at *δ* 6.88 (d, *J* = 8.5 Hz, 1H, H-5), 6.75 (overlap, 1H, H-6), 6.77 (overlap, 1H, H-8), 6.95 (d, *J* = 1.5 Hz, 1H, H-2′), 6.86 (overlap, 1H, H-7′), 6.96 (overlap, 1H, H-8′), 6.83 (d, *J* = 1.5 Hz, 1H, H-2″), 7.02 (overlap, 1H, H-5′), and 6.73 (overlap, 1H, H-6″). Four methine signals at *δ* 5.71 (t, *J* = 7.0, 2H, H-3, 5′) and 4.75 (t, *J* = 7.0 Hz, 2H, H-2, 4′); two methylene signals at *δ* 3.35 (t, *J* = 7.5 Hz, 2H, H-7b) and 2.70 (t, *J* = 7.0 Hz, 2H, H-7a); and three methyl proton signals at *δ* 1.90 (s, 3H), 1.89 (s, 3H), and 1.88 (s, 3H). The ^13^C-NMR data (Table 1), along with the DEPT spectrum, showed 30 carbon atoms, including three methyls at *δ* 22.6, 22.6, and 22.5 ppm; two methylene carbons at *δ* 42.1 and 35.8 ppm; four oxygenated and nine aromatic methine carbons; and 12 quaternary carbons (three carbonyl groups at *δ* 173.3 ppm and nine aromatic carbons). All these data pointed toward the unknown compound being a NADA polymer. Based on the analysis of spectral data, HR-ESI-MS data, and reported literature [11,12,13,14,15], it was concluded that compound **1** was a NADA trimer derivative. Therefore, the structure of **1** was constructed by analyzing HMBC and ^1^H-^1^H-COSY spectra (Figure 2). The ^1^H-^1^H-COSY correlations of H-7a/H-7b and H-6/H-5, as well as the HMBC correlations of H-7d/C-7c, H-7b/C-7c, H-7a/C-6, C-7, and C-8 revealed the position of the *N*-ethylformamide group at C-7. C-2/C-8a and C-3′/C-4′ were connected through an oxygen bridge. The proposed structure was supported by the HMBC correlations of H-6/C-4a, H-5/C-8a, H-2/C-8a, C-1′, and C-8′, H-2′/C-6′, H-7′/C-3′, H-4′/C-3′, C-1″, C-2″, C-6. Moreover, the HMBC correlations of H-3/C-3a and H-5′/C-5′a suggested that the acetyl groups were located at C-3 and C-5′. Therefore, the planar structure of compound **1** was determined with the aid of all HSQC, HMBC, and ^1^H-^1^H-COSY correlations (in Appendix A).

The coupling constants (*J* = 7.0 Hz) of H-2/H-3 and H-4′/H-5′ supported the finding that the relative configurations of C-2/C-3 and C-4′/C-5′ were all *trans* rather than *cis* (*J* = 1.5 Hz) [8,10]. In order to further clarify the absolute configuration of compound **1**, the optical rotation value and ECD spectrum were measured. It was found that compound **1** had no Cotton effect and the optical rotation value of **1** was zero, suggesting that **1** should be a racemic mixture. Thus, the enantiomers (+)-**1** and (−)-**1** were isolated using a chiral column. The experimental ECD curves of (+)-**1** and (−)-**1** were completely symmetrical by measuring. Consequently, after comparing the experimental ECD data with those that were quantum-mechanically calculated (Figure 3), enantiomers (+)-**1** and (−)-**1** were successively confirmed as 2*R*, 3*S*, 4′*R*, 5′*S* and 2*S*, 3*R*, 4′*S*, 5′*R* and named (±)-cryptamide A.

The ^1^H-NMR spectra of compounds **1**–**4** were almost identical, except for slight differences at *δ* 5.70 and 4.75 ppm, as shown in Figure 4. An analysis of NMR data for compounds **1** and **2** revealed that they have identical planar structures. Compounds **3** and **4** had the same molecular formula as that of compounds **1** and **2**, and their proton and carbon chemical shifts also closely resembled those of **1** and **2**, thus indicating that compounds **1**–**4** were all isomers. A careful examination of compounds **3** and **4** using NMR spectra revealed ^1^H-^1^H COSY correlations of H-6a/H-6b and H-7/H-8, as well as HMBC interactions of H-6d/C-6c, H-6b/C-6c, and H-6a/C-5, C-6 and C-7 (Figure 2). This indicated that the position of the *N*-ethylformamide group in compounds **3** and **4** is at C-6, which is different from compounds **1** and **2** (position C-7). This also confirmed that the planar structures of **3** and **4** were identical. The relative configurations at C-2/C-3 and C-4′/C-5′ of compounds **2**, **3**, and **4** were also determined to be *trans* by the coupling constants of H-2/H-3 and H-4′/H-5′ (*J* = 7.0 Hz). The optical rotation values and experimental ECD spectra of compounds **2**, **3**, and **4** also showed non-optical activity, indicating that they were all racemic mixtures. The enantiomers (+)-**2**/(−)-**2**, (+)-**3**/(−)-**3**, and (+)-**4**/(−)-**4** were separated on a chiral HPLC column. In a comparison of the experimental ECD spectra of compounds (±)-**2**, (±)-**3**, and (±)-**4** with those of calculated values (Figure 3), the absolute configurations of the (+)-**2** and (−)-**2** were successfully demonstrated to be 2*R*, 3*S*, 4′*R*, 5′*S* and 2*S*, 3*R*, 4′*S*, 5′*R*, named (±)-cryptamide B, while (+)-**3** and (−)-**3** were 2*R*, 3*S*, 4′*S*, 5′*R* and 2*S*, 3*R*, 4′*R*, 5′*S*, named (±)-cryptamide C, and (+)-**4** and (−)-**4** were 2*R*, 3*S*, 4′*R*, 5′*S* and 2*S*, 3*R*, 4′*S*, 5′*R*, named (±)-cryptamide D, respectively.

### 2.2. Nitric Oxide Inhibitory Activity

(±)-Cryptamides A–D were tested for their inhibitory effects on nitric oxide (NO) production in LPS-stimulated RAW264.7 macrophage cells. The viability of activated cells treated with (±)-cryptamides A–D at 25, 50, and 75 μmol/L was measured using the MTT method. When compared with the control group, (±)-cryptamide B, (±)-cryptamide C, and (−)-cryptamide D displayed no significant cellular toxicity at these concentrations for 24 h. Subsequently, the levels of NO released in RAW264.7 cells induced by LPS were measured using the Griess reagent. (±)-Cryptamide B, (±)-cryptamide C, and (−)-cryptamide D suppressed NO production in a dose-dependent manner (Figure 5), and (−)-cryptamide D had the strongest inhibition effect.

## 3. Materials and Methods

### 3.1. General Experimental Procedures

The Jasco P-1020 polarimeter (Jasco, Easton, MD, USA) was used to acquire optical rotations. UV spectra were acquired on an Agilent 8453 UV-visible spectrophotometer (Agilent Technologies, Santa Clara, CA, USA). Experimental ECD spectra in MeOH were acquired in a quartz cuvette of 1 mm optical path length on a Chirascan V100 spectropolarimeter (Applied Photophysics Ltd., London, UK). A Waters Xevo G2 QTOF mass spectrometer with a Synapt G2 HDMS quadrupole time-of-flight (TOF) mass spectrometer (Waters, Milford, MA, USA) was used to measure the HR-ESI-MS data. Both 1D and 2D-NMR spectra of the isolated compounds were obtained on a Bruker AVANCE III 500 MHz spectrometer (Bruker, Billerica, MA, USA). For semi-preparative, high-performance liquid chromatography (HPLC), a SEP SP-5030 Binary HPLC pump, equipped with a Sep UV300 photodiode array detector (SEP Corporation, Beijing, China), was used to isolate and purify the compounds. The chiral resolution was carried out using the Shimadzu Prominence HPLC system with SPD-M20A series Prominence HPLC UV-Vis detectors (Shimadzu, Tokyo, Japan) equipped with a Chiral-INA (250 × 4.6 mm i.d., 5 μm) column (Guangzhou FLM Scientific Instrument Co., Guangzhou, China). An LC/MS analysis was carried out using an Agilent 1200 series HPLC system (Agilent Technologies, Santa Clara, CA, USA) equipped with a diode array detector and a 6130 series ESI mass spectrometer, using an analytical Waters column (2.1 × 50 mm, 1.7 μm). Cosmosil RP-C18 silica gel (Nacalai Tesque, Inc., Kyoto, Japan), MCI gel CHP 20P (75–150 μm, Mitsubishi Chemical Industries, Tokyo, Japan), HW-40F (Tosoh Corporation, Tokyo, Japan), and Sephadex LH-20 (GE Healthcare, Uppsala, Sweden) were used for column chromatography (CC). Silica gel HSGF254 plates were used for thin-layer chromatography (TLC). Spots on the TLC were detected under UV light or by using iodine.

### 3.2. Plant Material

The dried Periostracum cicadae was purchased from Zhengzhou medicine Co., Ltd., Zhengzhou (Henan), PR China, in September 2020, and was identified by Professor Guo Tao of Henan University of Traditional Chinese Medicine as the skin shell shed from the insect *Cryptotympana pustulata* Fabricius.

### 3.3. Extraction and Isolation

The powder of the dried Periostracum cicadae (5 kg) was extracted with 70% ethanol for 72 h each time, for a total of three times. The crude extract (631 g) was obtained through concentrating the extract under pressure. This extract was loaded onto a macroporous resin column and eluted with MeOH-H_2_O (10:90–100:0, *v*/*v*, gradient system) to provide five fractions (A–E). 

Fraction C (21 g) was divided into five parts (Fr.C1-C4) by using an MCI gel CHP 20P column eluted with gradient aqueous MeOH (10:90–100:0, *v*/*v*, gradient system). Compounds **10** (4.9 mg) and **11** (7.6 mg) were obtained through silica gel CC with a gradient solvent system of aqueous MeOH (60:40–100:0, *v*/*v*) from Fr.C2. 

Fraction D (134 g) was divided into five parts (Fr.D1-D5) by using an MCI gel CHP 20P column eluted with gradient aqueous MeOH (10:90–100:0, *v*/*v*, gradient system). Fraction D1 was chromatographed by RP-C_18_ gel (aqueous MeOH, 40:60–100:0, *v*/*v*), Sephadex LH-20 (MeOH), and RP-C_18_ semi-preparative HPLC (35% MeOH/H_2_O, 3 mL/min, ES Industries C_18_ (ES Industries, INC., West Berlin, NJ, USA), 250 × 10.0 mm i.d., 5 μm), ultimately leading to the purification of compounds **5** (10.1 mg, t_R_ 13 min), **6** (7.9 mg, t_R_ 19 min), and **7** (12.4 mg, t_R_ 33 min). Fraction D2 (23.3 g) was submitted to the RP-C_18_ gel column (aqueous MeOH, 40:60–100:0) to produce six fractions (Fr.D2.1–Fr.D2.6). Fr.D2.1 was purified using semi-preparative HPLC with 40% CH_3_CN/H_2_O mixtures (3 mL/min) to develop compounds **12** (9.8 mg, t_R_ 38 min) and **13** (13.4 mg, t_R_ 41 min). Fraction Fr.D2.3 was gel filtrated over a Sephadex LH-20 (MeOH) column to obtain two subfractions (A1 and A2). A1 (166 mg) was further separated using the semi-preparative high-performance liquid phase (HPLC) (ES Industries C_18_, 250 × 10.0 mm i.d., 5 μm) with 20% CH_3_CN/H_2_O (isocratic system, flow rate: 3 mL/ min) to yield compounds **1** (20.4 mg, t_R_ 43.5 min), **2** (11.0 mg, t_R_ 47.4 min), **3** (26.8 mg, t_R_ 57.0 min), and **4** (12.5 mg, t_R_ 62.8 min). Then, the separation of the enantiomers was carried out using a Phenomenex Chiral-INA (250 × 4.6 mm i.d., 5 μm) column. Ultimately, the enantiomers (+)-**1** (1.2 mg, t_R_ 24.1 min) and (−)-**1** (1.1 mg, t_R_ 35.0 min) were isolated using a chiral column with *n*-hexane/ethanol (4:1, *v*/*v*, isocratic system, flow rate: 1 mL/min). Similar to the procedure used for **1**, the other enantiomers were isolated using the same chiral HPLC column: (+)-**2** (0.9 mg, t_R_ 11.5 min) and (−)-**2** (1.1 mg, t_R_ 18.0 min), (+)-**3** (4.6 mg, t_R_ 21.0 min) and (−)-**3** (1.8 mg, t_R_ 32.5 min), and (+)-**4** (2.8 mg, t_R_ 23.6 min) and (−)-**4** (1.2 mg, t_R_ 37.0 min). Compounds **8** (22.9 mg, t_R_ 16 min), **9** (18.5 mg, t_R_ 27 min), and **14** (15.5 mg, t_R_ 34 min) were obtained through the subfractions of A2 with the aid of a semi-preparative HPLC (25% CH_3_CN/H_2_O, 3 mL/min, ES Industries C_18_, 250 × 10.0 mm i.d., 5 μm). 

### 3.4. Compound Characterization

(±)Cryptamide A (**1**): white powder; [α]^D^_20_ + 70 (c 0.1, MeOH) for (+)-**1** and [α]^D^_20_ –132.5 (c 0.1, MeOH) for (−)-**1**; UV (MeOH) λmax (log ε) 283 (0.038), 206 (0.503), and 250 (−0.014) nm; ECD (MeOH) λmax (△ε) 199 (+3.41), 207 (+2.97), 231 (−3.20), and 288 (−0.94) nm for (+)-**1** and ECD (MeOH) λmax (△ε) 199 (–2.99), 207 (−3.06), 231 (+1.55), and 288 (+0.10) nm for (−)-**1**; ^1^H (500 MHz) and ^13^C-NMR (125 MHz), see Table 1; positive HR-ESI-MS *m/z* 578.2139 [M + H]^+^ (calculated for C_30_H_31_N_3_O_9_H).

(±)-Cryptamide B (**2**): white powder; [α]^D^_20_ + 15 (c 0.1, MeOH) for (+)-**2** and [α]^D^_20_ –35 (c 0.1, MeOH) for (−)-**2**; UV (MeOH) λmax (log ε) 283 (0.053), 207 (0.652), and 249 (−0.008) nm; ECD (MeOH) λmax (△ε) 200 (+3.97), 211 (−6.71), 240 (+0.85), and 291 (+0.45)nm for (+)-**2** and ECD (MeOH) λmax (△ε) 200 (–6.45), 211 (+8.89), 240 (−1.26), and291 (−0.55) nm for (−)-**2**; ^1^H (500 MHz) and ^13^C-NMR (125 MHz), see Table 1; positive HR-ESI-MS *m/z* 578.2136 [M + H]^+^ (calculated for C_30_H_31_N_3_O_9_H).

(±)Cryptamide C (**3**): white powder; [α]^D^_20_ + 280 (c 0.1, MeOH) for (+)-**3** and [α]^D^_20_ –270 (c 0.1, MeOH) for (−)-**3**; UV (MeOH) λmax (log ε) 283 (0.068), 206 (0.821), and 249 (−0.009) nm; ECD (MeOH) λmax (△ε) 194 (+16.78), 208 (−5.73), 231 (−9.10), and 286 (−1.83)nm for (+)-**3** and ECD (MeOH) λmax (△ε) 194 (–15.73), 208 (+4.68), 231 (+7.42), and 286 (+0.93) nm for (−)-**3**; ^1^H (500 MHz) and ^13^C-NMR (125 MHz), see Table 1; positive HR-ESI-MS *m/z* 578.2131 [M + H]^+^ (calculated for C_30_H_31_N_3_O_9_H).

(±)Cryptamide D (**4**): white powder; [α]^D^_20_ + 32.5 (c 0.1, MeOH) for (+)-**4** and [α]^D^_20_ –60 (c 0.1, MeOH) for (−)-**4**; UV (MeOH) λmax (log ε) 283 (0.033), 206 (0.394), and 249 (−0.009) nm; ECD (MeOH) λmax (△ε) 201 (+1.09), 215 (+0.22), 239 (+0.49), and 288 (+0.28) nm for (+)-**4** and ECD (MeOH) λmax (△ε) 201 (–1.92), 215 (−1.71), 238 (−1.31), and 288 (−0.72) nm for (−)-**4**; ^1^H (500 MHz) and ^13^C-NMR (125 MHz), see Table 1; positive HR-ESI-MS *m/z* 578.2137 [M + H]^+^ (calculated for C_30_H_31_N_3_O_9_H).

### 3.5. Computational Analysis

Conformation searches for compounds were performed by Spartan’s 14 software (Wavefunction, Inc., Irvine, CA, USA) using Merck Molecular Force Field (MMFF) level. Density-functional theory (DFT) optimization for the low-energy conformations of compounds **1**–**4** (in Appendix A) was carried out at a b3lyp/6–31 g (d, *p*) level using the PCM solvation model with methanol represented by a dielectric constant. The optimized structures were subjected to the frequency calculations at the b3lyp/6–31 g (d, *p*) level to determine the true minimum energy position and generate thermodynamic data. The optimized structures were further calculated using time-dependent density functional theory (TDDFT) under b3lyp/6–31 g (d, *p*). The rotational intensities of 80 excited states were calculated. SpecDis 1.53 and GraphPad Prism 5 were used to generate ECD spectra based on dipole length rotation intensity using Gaussian band shapes with =0.3 eV.

### 3.6. Cell Culture and Nitric Oxide Inhibitory Assay

RAW264.7 cells, provided by Professor Chen Suiqing (Henan University of Chinese Medicine, China), were cultured using high-glucose Dulbecco’s Modified Eagle Medium (DMEM, WH01122009XP02, Procell, Wuhan, China) containing 10% fetal bovine serum (FBS, 42G6185K, Gibco, Big Cabin, OK, USA), 100 U/mL penicillin, and 100 μg/mL streptomycin (Solarbio, China) at 37 °C in a humidified carbon dioxide incubator with 5% CO_2_.

The cell viability was determined through the MTT method. After the logarithmic growth phase cells were digested and resuspended, the cell density was adjusted to 4 × 10^4^/mL and 200 μL of solution per well was incubated in a 96-well plate for 24 h. Except for the blank control group, the RAW264.7 cells were treated with a compounds culture medium containing different concentrations (25 μM, 50 μM, and 75 μM) for 24 h. Then, 20 μL of thiazolyl blue tetrazolium bromide (MTT, M8180, Solarbio, Beijing, China) solution (5 mg/mL) was added to each hole of the 96-well plate and incubated at 37 °C for 4 h. Subsequently, the absorbance value at the wavelength of 492 nm was recorded using a microplate reader (Tecan, Mannedorf, Switzerland).

After that, the RAW264.7 cells (4 × 10^4^ cells/well) were inoculated in 96-well plates overnight. Then, the cells were incubated with the presence or absence of compounds for 1 h, and were incubated with the presence or absence of LPS for another 24 h. At the end of the incubation, the cell culture supernatant was collected to detect the nitrite content of different groups. Then, the release of NO was determined using a commercial NO assay kit (S0021, Beyotime Institute of Biotechnology, Shanghai, China).

### 3.7. Statistical Analysis

All experimental data were expressed as mean ± standard deviation (SD) values. The normality was tested using the Shapiro-Wilk method. The data conforming to a normal distribution were compared among multiple groups (using one-way ANOVAs). Statistical differences were estimated using SPSS 25.0, and analysis results at *p* < 0.05 were considered statistically significantly different. Data represent the means ± SEM for *n* = 6.

## 4. Conclusions

In summary, the chemical investigation of PC led to the isolation of four pairs of novel dopamine enantiomer trimers (**1**–**4**) and 10 pairs of dopamine enantiomer dimers (**5**–**14**). The structures of new compounds were confirmed by one-dimensional and two-dimensional NMR spectra, HR-ESI-MS data, and ECD computational methods. In addition, all the new compounds were evaluated for their inhibitory effects against NO production induced by LPS in RAW264.7 cells. Among them, (±)-cryptamide B (**2**), (±)-cryptamide C (**3**), and (−)-cryptamide D (**4**) exhibited significant inhibitory effects against NO production, which could be useful in the development of new potential anti-inflammatory agents.

## Figures and Tables

**Figure 1 molecules-27-06707-f001:**
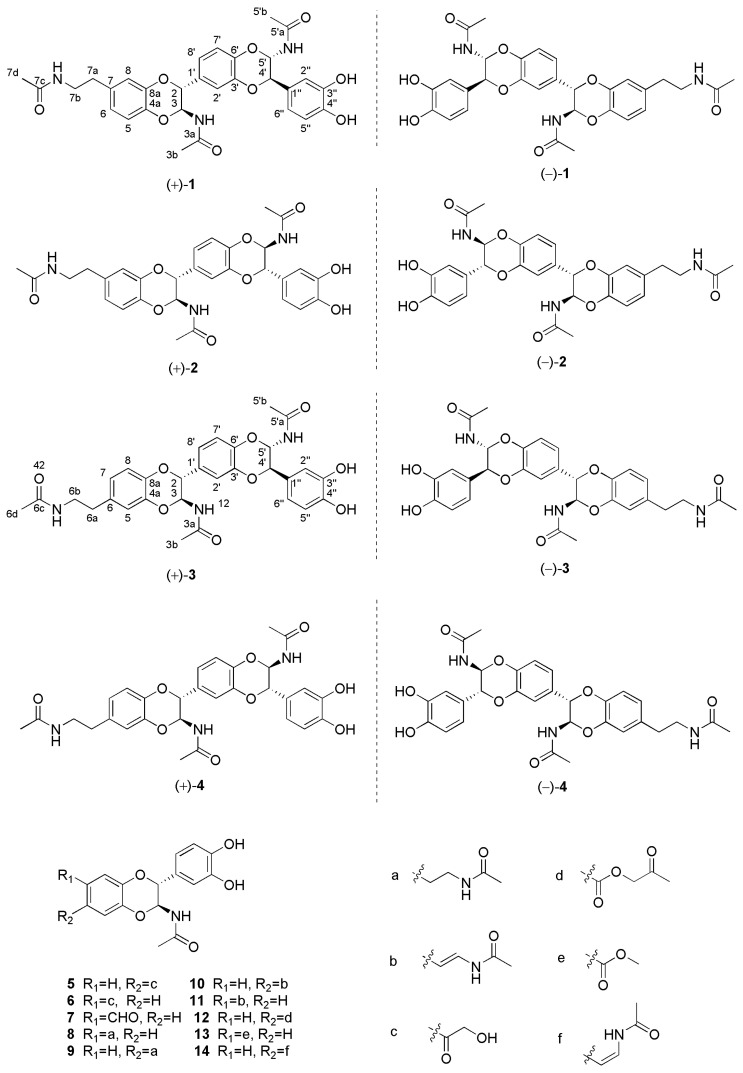
The chemical structures of the isolated compounds **1**–**14**.

**Figure 2 molecules-27-06707-f002:**
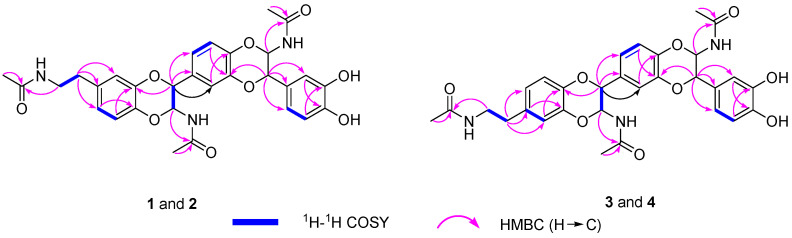
Key correlations of ^1^H-^1^H-COSY and HMBC for compounds **1**–**4**.

**Figure 3 molecules-27-06707-f003:**
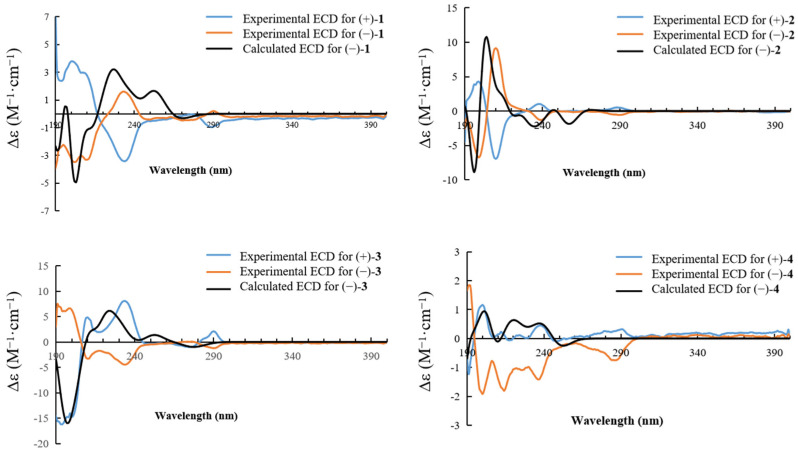
ECD spectra of compounds **1**, **2**, **3**, and **4** in MeOH.

**Figure 4 molecules-27-06707-f004:**
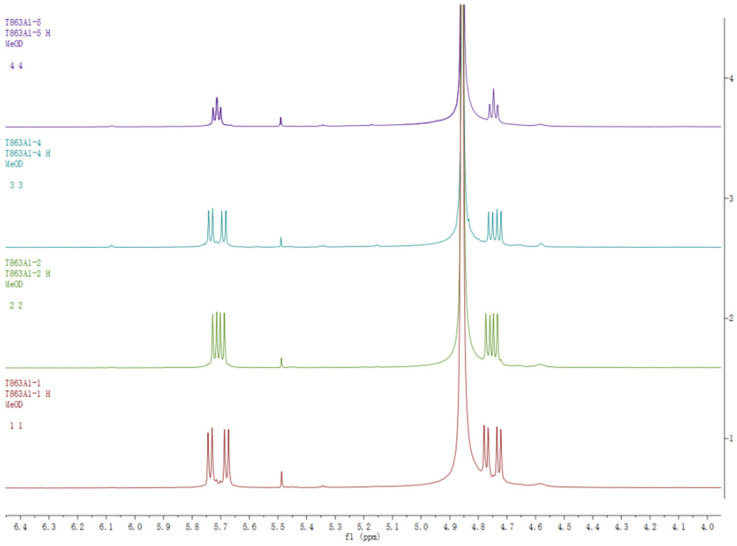
^1^H-NMR comparison of compounds **1**, **2**, **3**, and **4**.

**Figure 5 molecules-27-06707-f005:**
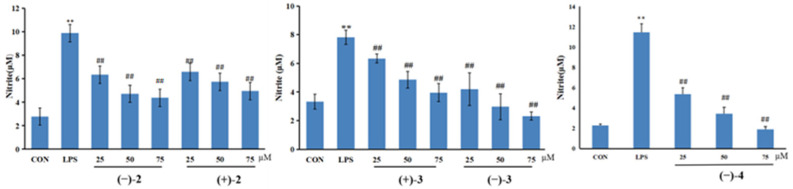
RAW264.7 cells were seeded in a 96-well plate and cultured with different concentrations (25, 50, and 75 μM) of (±)-**2**, (±)-**3**, and (−)-**4** for 1 h before stimulation with LPS (1 μg/mL) for 24 h. The Griess reagent was used to quantify the nitrite level in the cell supernatant. Data are represented as mean ± SEM (*n* = 6). ** *p* < 0.01 indicates statistically significant difference when compared with control group; ^##^ *p* < 0.01 indicating statistical.

**Table 1 molecules-27-06707-t001:** ^1^H-NMR and ^13^C-NMR Data of Compounds **1**, **2**, **3**, and **4**.

No.	1		2		3		4	
	*δ*_C_^a^/ppm	*δ*_H_/ppm (^b^ *J* in Hz)	*δ*_C_/ppm	*δ*_H_/ppm (*J* in Hz)	*δ*_C_/ppm	*δ*_H_/ppm (*J* in Hz)	*δ*_C_/ppm	*δ*_H_/ppm (*J* in Hz)
2	78.3	4.75 (t, 7.0)	78.2	4.77 (d, 7.0)	78.3	4.77 (d, 7.0)	78.3	4.76 (d, 7.0)
3	78.4	5.71 (t, 7.0)	78.3	5.74 (d, 7.0)	78.4	5.72 (d, 7.0)	78.3	5.74 (d, 7.0)
4a	142.8		142.1		142.1		142.8	
5	118.0	6.88 (d, 8.5)	118.1	6.87 (overlap)	118.0	6.77 (overlap)	118.0	6.77 (overlap)
6	123.2	6.75 (overlap)	123.4	6.75 (overlap)	134.3		134.5	
7	134.5		134.3		123.4	6.75 (overlap)	123.2	6.75 (overlap)
8	118.2	6.77 (overlap)	118.1	6.77 (overlap)	118.1	6.83 (overlap)	118.2	6.84 (overlap)
8a	144.5		144.4		144.3		144.4	
1′	131.0		131.0		130.1		131.0	
2′	117.9	6.95 (d, 1.5)	118.0	6.95 (d, 1.5)	118.0	6.95 (d, 1.5)	117.9	6.95 (d, 1.5)
3′	144.3		144.2		144.1		144.2	
4′	77.9	4.75 (d, 7.0)	77.9	4.73 (d, 7.0)	78.0	4.74 (d, 7.0)	77.8	4.73 (d, 7.0)
5′	78.3	5.71 (d, 7.0)	78.2	5.68 (d, 7.0)	78.2	5.70 (d, 7.0)	78.3	5.69 (d, 7.0)
6′	143.5		144.1		144.4		143.5	
7′	117.3	6.86 (overlap)	117.4	6.82 (overlap)	117.3	6.82 (overlap)	117.3	6.84 (overlap)
8′	122.4	6.96 (overlap)	122.2	6.96 (overlap)	122.4	6.96 (overlap)	122.2	6.96 (overlap)
1″	128.6		128.5		128.5		128.6	
2″	115.5	6.83 (d, 1.5)	115.6	6.80 (s)	115.5	6.81 (s)	115.5	6.84 (d, 1.5)
3″	146.5		146.5		146.5		146.5	
4″	147.2		147.2		147.2		147.3	
5″	116.2	7.02 (overlap)	116.1	7.01 (overlap)	116.2	7.02 (overlap)	116.1	7.01 (overlap)
6″	120.6	6.73 (overlap)	120.6	6.73 (overlap)	120.6	6.73 (overlap)	120.6	6.73 (overlap)
3a	173.3		173.3		173.3		173.3	
3b	22.6	1.90 (s)	22.6	1.90 (s)	22.6	1.90 (s)	22.6	1.90 (s)
5′a	173.3		173.3		173.3		173.3	
5′b	22.6	1.89 (s)	22.6	1.89 (s)	22.6	1.89 (s)	22.6	1.89 (s)
7a/6a	35.8	2.70 (t, 7.0)	35.8	2.70 (t, 7.0)	35.8	2.70 (t, 7.0)	35.8	2.70 (t, 7.0)
7b/6b	42.1	3.35 (t, 7.5)	42.1	3.35 (t, 7.5)	42.1	3.35 (t, 7.5)	42.1	3.35 (t, 7.5)
7c/6c	173.3		173.2		173.2		173.2	
7d/6d	22.5	1.88 (s)	22.5	1.88 (s)	22.5	1.88 (s)	22.5	1.88 (s)

^a 13^C-NMR data were assigned based on HSQC and HMBC experiments. ^b^ Coupling constants (in parentheses) are in Hz.

## Data Availability

Not applicable.

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
