# Peer review of "(±)-Cryptamides A–D, Four Pairs of Novel Dopamine Enantiomer Trimers from the Periostracum Cicadae"

_molecules, 2022, doi:10.3390/molecules27196707_

Round 1
Reviewer 1 Report
The paper entitled „(±)-Cryptamides A-D, four pairs of novel dopamine enantiomer 2 trimers from the Periostracum cicadae” by Feng et al reports the discovery of four enantiomeric pairs of novel dopamine trimers. It also mentions the identification of further 10 enantiomeric pairs of dopamine dimers which were also isolated from the same source, but were already described in literature.
The research seems to be performed methodically and all new molecules were characterised well, using NMR spectroscopy, HRMS and ECD computational methods.
The whole article needs rewording and English corrections, please consult a native speaker to go through the text and make corrections before resubmitting.
δ is the symbol for chemical shift which has the unit “ppm”. The unit is missing throughout the whole manuscript, including the Table 1. Please correct.
SciFinder search revealed two references with trimers of similar structure, they should be commented.
1. Lee, Seoung Rak; Yi, Sang Ah; Nam, Ki Hong; Park, Jae Gyu; Hwang, Jae Sam; Lee, Jaecheol; Kim, Ki Hyun “(±)-Kituramides A and B, pairs of enantiomeric dopamine dimers from the two-spotted cricket Gryllus bimaculatus” Bioorganic Chemistry (2020), 95, 103554 | Language: English, Database: CAplus and MEDLINE
2. Patent: Li, Guoyu; Yang, Lu; Lu, Lina; Yu, Fei; Wang, Jinhui Dopamine polymer and its derivative, and preparation method and medical application thereof” China, CN102285958 A 2011-12-21 | Language: Chinese, Database: CAplus
3.
Supplementary information is comprehensive, it includes all the NMR and HRMS spectra on the novel compounds. However, due to the large overlap, the zoomed aromatic region of HSQC and HMBC spectra should be included after the figure of each HSQC and HMBC spectrum.
1. Abstract:
Four pairs of novel dopamine enantiomer trimers, (±)-cryptamides A-D (1-4), and 10 pairs of previously described dopamine enantiomer dimers (5-14) were isolated from the Periostracum cicadae, the cast-off shell of insect Cryptotympana pustulata. Besides being pairs of enantiomers, the eight trimers were also elucidated to be regioisomers, most probably resulting from their mechanism of formation, [4+2] cycloaddition. The discovery of dopamine trimers is rarely reported when it comes to natural products derived from insects.
2. Introduction
a) Line 26: Periostracum cicadae (PC), the cast-off shell of insect Cryptotympana pustulata Fabricius, is a traditional Chinese medicine called “chantui” in China and mainly produced in Shandong, Henan, Hebei, Hubei, Jiangsu and Sichuan provinces [1].
b) Line 27: PC has the functions of evacuating wind heat, benefiting pharynx, penetrating rash, brightening eyes, removing pannus and relieving spasm, which is often used as heat-clearing medicine in clinic [2].
This sentence needs to be rewritten or removed. It should start with “PC has been used in traditional Chinese medicine for…” and then the proper names for ailments should be used.
c) Line 30: Pharmacological studies of PC in previous led to the demonstration of anti-convulsion, anti-inflammation, anti-tumor, antitussive, expectorant, antiasthmatic, immuno-suppression and anti-allergy.
This sentence needs to be rewritten. Suggestion: Previously described pharmacological studies demonstrate anti-convulsion, anti-inflmmatory, anti-tumor, antitussive, expectorant, anti-asthmatic, immune-supressive and anti-allergic properties of PC [3-6].
d) Line 32: Chemical investigations showed that the main small molecular components contained in PC were N-acetyldopamine (NADA) derivatives [2,5-10].
This sentence needs to be rewritten. Suggestion: Previously reported chemical investigations showed that the most common molecular components isolated from PC were N-acetyldopamine (NADA) derivatives [2,5-10].
e) Line 34: It should be mentioned that the dimers and tetramers of NADA were the richest in PC, but the discovery of trimers was rarely reported.
This sentence needs to be rewritten. Suggestion: At this point it should be mentioned that most studies report findings of NADA dimers and tetramers, while the discovery of trimers is unusual.
f) Line 36: This study reported the isolation and structural identification of four pairs of novel dopamine enantiomer trimers (1-4) (Figure 1) and 10 pairs of dopamine enantiomer dimers from PC. A detailed comparison indicated that these four pairs of compounds were all isomers.
This sentence needs to be rewritten. Suggestion: Hereby we report the isolation and structural elucidation of four novel enantiomeric dopamine trimer pairs (1-4) (Figure 1) and 10 enantiomeric dopamine dimer pairs, the latter already described in literature (put references here). A detailed comparison revealed that the novel four pairs of compounds were also regioisomers.
3. Results
a) Figure 1, line 47: please put numbers on the structure 3 (as you did on structure 1), and check if the chemical shifts of these numbers are correct in the Table 1. Table states that 13C chemical shift of C-6 in all compounds is 123 ppm and that it is CH with proton chemical shift of 6.96 ppm, which is only so if the numbering of compounds 3 and 4 is different than 1 and 2.
b) Line 61: HR-ESI-MS
c) Line 75, 78, 80, 82, 108 and 109: instead of “corrections” should be HMBC and COSY “correlations” or “interactions”
d) Line 71: All these data were similar to those of NADA polymers.
Suggestion: All these data were pointing towards the unknown compound to be NADA polymer.
e) Line 73: instead of “speculated” use “concluded”
f) Line 73: instead of “in addition” use “therefore”
g) Line 78: “respectively” is not needed here
h) Line 78: instead “this assignment” use “the proposed structure”
i) Line 81: instead of “finally” use “therefore”
j) Line 87: is there a cis relative configuration described in literature? If yes, it should be mentioned here and the size of coupling constant characteristic for cis.
k) Line 92: instead of “by” use “using”
l) Line 100: instead “the same” use “identical”
m) Line 100: “the slight difference in signals at δ 5.70 and 4.75 ppm” – unit for chemical shift is “ppm”, so when giving a number, a unit should be written.
n) Line 101: “Compounds 2 and 1 shared the same planar structure by analyzing their 1D and 2D NMR data” Suggestion: “Analysis of NMR data for compounds 1 and 2 revealed that they have identical planar structure.”
o) Line 103: “1H and 13C spectroscopic values” should be changed to “proton and carbon chemical shifts”
p) Line 105: A careful comparison of NMR spectra of compounds 3 and 4 with compounds 1 and 2 revealed that the position of the N-ethylformamide group located at C-6 in 3 and 4, which was different from compounds 1 and 2. This inference was further confirmed through the HMBC corrections of H-6d/C-6c, H-6b/C-6c, and H-6a/C-5, C-6, and C-7 along with the 1H1H COSY corrections of H-6a/H-6b and H-6/H-5 (Figure 2). Thus, the planar structures of 3 and 4 were ensured to be the same.
Suggestion: Careful examination of compounds 3 and 4 NMR spectra revealed COSY correlations of H-6a/H-6b and H-6/H-5, as well as HMBC interactions H-6d/C-6c, H-6b/C-6c, and H-6a/C-5, C-6, and C-7 (Figure 2). This indicated that the position of N-ethylformamide group in compounds 3 and 4 is at C-6, which is different to compounds 1 and 2 (position C-7), but also confirmed that the planar structures of 3 and 4 were identical.
4. Supplementary Information
a) The zoomed aromatic region of HSQC and HMBC spectra should be included after the figure of each HSQC and HMBC spectrum.
b) HSQC spectrum should be phase sensitive, but on the figure all the peaks are the same colour? Why is that?
Author Response
Response to Reviewer 1 Comments
1. The whole article needs rewording and English corrections, please consult a native speaker to go through the text and make corrections before resubmitting.
Answer: Thanks for your good suggestion. The whole article has been rewritten and corrected for English spelling. Additionally, our manuscript has been improved by Professor Yi Zhao, working in Department of Biological Sciences, Lehman College, The City University of New York. She also has been focusing on natural products research.
2. δ is the symbol for chemical shift which has the unit “ppm”. The unit is missing throughout the whole manuscript, including the Table 1. Please correct.
Answer: we have added the unit “ppm” of chemical shift in table 1. However, the unit “ppm” was not added in compound structure identification section in manuscript text according to the published articles in molecules.
3. SciFinder search revealed two references with trimers of similar structure, they should be commented.
1. Lee, Seoung Rak; Yi, Sang Ah; Nam, Ki Hong; Park, Jae Gyu; Hwang, Jae Sam; Lee, Jaecheol; Kim, Ki Hyun “(±)-Kituramides A and B, pairs of enantiomeric dopamine dimers from the two-spotted cricket Gryllus bimaculatus” Bioorganic Chemistry (2020), 95, 103554 | Language: English, Database: CAplus and MEDLINE
2. Patent: Li, Guoyu; Yang, Lu; Lu, Lina; Yu, Fei; Wang, Jinhui Dopamine polymer and its derivative, and preparation method and medical application thereof” China, CN102285958 A 2011-12-21 | Language: Chinese, Database: CAplus
Answer:Thanks for your good suggestion. The trimers in reference 1 (Bioorganic Chemistry (2020), 95, 103554) did not provided any data. It only has a structure that was cited by the reference 2 (CN102285958 A 2011-12-21) you provide. For reference 2 (CN102285958 A 2011-12-21), it is a Chinese patent application, not a formal professional journal. There are no NMR, UV, ECD, HR-ESI-MS, Optical rotation data for trimers in this reference 2, which are not persuasive.
4. Supplementary information is comprehensive, it includes all the NMR and HRMS spectra on the novel compounds. However, due to the large overlap, the zoomed aromatic region of HSQC and HMBC spectra should be included after the figure of each HSQC and HMBC spectrum.
Answer: Thanks for your good suggestion. The zoomed aromatic region of HSQC and HMBC spectra have been included after the figure of each HSQC and HMBC spectrum in Supplementary information.
1. Abstract
Answer: The abstract of the paper has been changed to “Four pairs of novel dopamine enantiomer trimers, (±)-cryptamides A-D (1-4), and 10 pairs of previously described dopamine enantiomer dimers (5-14) were isolated from the Periostracum cicadae, the cast-off shell of insect Cryptotympana pustulata. Besides being pairs of enantiomers, the eight trimers were also elucidated to be regioisomers, most probably resulting from their mechanism of formation, [4+2] cycloaddition. The discovery of dopamine trimers is rarely reported when it comes to natural products derived from insects.” in our revised manuscript according to your suggestion.
2. Introductiona) Line 26: Periostracum cicadae (PC), the cast-off shell of insect Cryptotympana pustulata Fabricius, is a traditional Chinese medicine called “chantui” in China and mainly produced in Shandong, Henan, Hebei, Hubei, Jiangsu and Sichuan provinces [1].
Answer: We have deleted “in China” in this sentence according to your suggestion.
b) Line 27: PC has the functions of evacuating wind heat, benefiting pharynx, penetrating rash, brightening eyes, removing pannus and relieving spasm, which is often used as heat-clearing medicine in clinic [2]. This sentence needs to be rewritten or removed. It should start with “PC has been used in traditional Chinese medicine for…” and then the proper names for ailments should be used.
Answer: Thank you for your good suggestion. This sentence has been changed to “PC has been used in traditional Chinese medicine for heat-clearing medicine in clinic [2]”.
c) Line 30: Pharmacological studies of PC in previous led to the demonstration of anticonvulsion, anti-inflammation, anti-tumor, antitussive, expectorant, antiasthmatic, immunosuppression and anti-allergy.This sentence needs to be rewritten. Suggestion: Previously described pharmacological studies demonstrate anti-convulsion, anti-inflmmatory, anti-tumor, antitussive, expectorant, anti-asthmatic, immune-supressive and anti-allergic properties of PC [3-6].
Answer: According to your good suggestion, this sentence has been revised to “Previously described pharmacological studies demonstrate anti-convulsion, anti-inflmmatory, anti-tumor, antitussive, expectorant, anti-asthmatic, immune-supressive and anti-allergic properties of PC [3-6].”
d) Line 32: Chemical investigations showed that the main small molecular components contained in PC were N-acetyldopamine (NADA) derivatives [2,5-10].This sentence needs to be rewritten. Suggestion: Previously reported chemical investigations showed that the most common molecular components isolated from PC were N-acetyldopamine (NADA) derivatives [2,5-10].
Answer: According to your good suggestion, this sentence has been changed to “Previously reported chemical investigations show that the most common molecular components isolated from PC are N-acetyldopamine (NADA) derivatives [2,5-10].”
e) Line 34: It should be mentioned that the dimers and tetramers of NADA were the richest in PC, but the discovery of trimers was rarely reported.This sentence needs to be rewritten. Suggestion: At this point it should be mentioned that most studies report findings of NADA dimers and tetramers, while the discovery of trimers is unusual.
Answer: According to your good suggestion, this sentence has been changed to “At this point it should be mentioned that most studies report findings of NADA dimers and tetramers, while the discovery of trimers is unusual.”
f) Line 36: This study reported the isolation and structural identification of four pairs of novel dopamine enantiomer trimers (1-4) (Figure 1) and 10 pairs of dopamine enantiomer dimers from PC. A detailed comparison indicated that these four pairs of compounds were all isomers.This sentence needs to be rewritten. Suggestion: Hereby we report the isolation and structural elucidation of four novel enantiomeric dopamine trimer pairs (1-4) (Figure 1) and 10 enantiomericdopamine dimer pairs, the latter already described in literature (put references here). A detailed comparison revealed that the novel four pairs of compounds were also regioisomers.
Answer: Thank you for your good suggestion. We have revised this sentence according to your suggestion.
3. Results
a) Figure 1, line 47: please put numbers on the structure 3 (as you did on structure 1), and check if the chemical shifts of these numbers are correct in the Table 1. Table states that 13C chemical shift of C-6 in all compounds is 123 ppm and that it is CH with proton chemical shift of 6.96 ppm, which is only so if the numbering of compounds 3 and 4 is different than 1 and 2.
Answer: We are so sorry for our mistakes. We have put numbers on the structure 3. Additionally, we carefully checked the chemical shifts of all compounds and corrected them in table 1. The 13C chemical shifts of C-6 in all compounds are different, which has been revised.
b) Line 61: HR-ESI-MS
Answer: According to your good suggestion, “HRESIMS” (Line 61) has been changed as “HR-ESI-MS”
c) Line 75, 78, 80, 82, 108 and 109: instead of “corrections” should be HMBC and COSY “correlations”
Answer: we are so sorry for our spelling error. The “ corrections” are replaced by “correlations” in our revised manuscript.
d) Line 71: All these data were similar to those of NADA polymers.
Answer: Thanks for your good suggestion. We have changed these sentence in our revised manuscript as “All these data were pointing towards the unknown compound to be NADA polymer.”
e) Line 73: instead of “speculated” use “concluded”
f ) Line 73: instead of “in addition” use “therefore”
g) Line 78: “respectively” is not needed here
h) Line 78: instead “this assignment” use “the proposed structure”
i) Line 81: instead of “finally” use “therefore”
Answer: Thanks for your good suggestion. The word and phrase ( line 73, 78 and 81) have been changed as you suggested, and “respectively”( line 78) has been removed in our revised manuscript.
J) Line 87: is there a cis relative configuration described in literature? If yes, it should be mentioned here and the size of coupling constant characteristic for cis.
Answer: In reference 10, there is a cis relative configuration described. We have added the size of coupling constant characteristic for cis and reference in our revised manuscript. “The coupling constants (J = 7.0 Hz) of H-2/H-3 and H-4’/H-5’ supported the finding that the relative configurations of C-2/C-3 and C-4’/C-5’ were all trans rather than cis (J = 1.5 Hz) [8,10].” k) Line 92: instead of “by” use “using”
l) Line 100: instead “the same” use “identical”
Answer: Thanks for your good suggestion. According to your good suggestion, “by” has been changed as “using” and “the same” (Line 100) are replaced by “identical” in our revised manuscript.
- m) Line 100: “the slight difference in signals at δ70 and 4.75 ppm” – unit for chemical shift is “ppm”, so when giving a number, a unit should be written.
Answer: Thank you for your good advice. We have changed this sentence in our revised manuscript as follows: “the slight difference in signals at δ 5.70 and 4.75 ppm.
- n) Line 101: “Compounds 2 and 1 shared the same planar structure by analyzing their 1D and 2D NMR data” Suggestion: “Analysis of NMR data for compounds 1 and 2 revealed that they have identical planar structure.”
Answer: According to your good suggestion, this sentence has been changed in our revised manuscript.
- o) Line 103: “1H and 13C spectroscopic values” should be changed to “proton and carbon chemical shifts”
- p) Line 105: A careful comparison of NMR spectra of compounds 3 and 4 with compounds 1 and 2 revealed that the position of the N-ethylformamide group located at C-6 in 3 and 4, which was different from compounds 1 and 2. This inference was further confirmed through the HMBC corrections of H-6d/C-6c, H-6b/C-6c, and H-6a/C-5, C-6, and C-7 along with the 1H-1H COSY corrections of H-6a/H-6b and H-6/H-5 (Figure 2). Thus, the planar structures of 3 and 4 were ensured to be the same.
Answer: According to your good suggestion, we have changed these sentences as follows in our revised manuscript.
- o) Line 103: proton and carbon chemical shifts.
- p) Line 105: Careful examination of compounds 3 and 4 NMR spectra revealed COSY correlations of H-6a/H-6b and H-6/H-5, as well as HMBC interactions H-6d/C-6c, H-6b/C-6c, and H-6a/C-5, C-6, and C-7 (Figure 2). This indicated that the position of N-ethylformamide group in compounds 3 and 4 is at C-6, which is different to compounds 1 and 2 (position C-7), but also confirmed that the planar structures of 3 and 4 were identical.
4. Supplementary Information
a) The zoomed aromatic region of HSQC and HMBC spectra should be included after the figure of each HSQC and HMBC spectrum. b) HSQC spectrum should be phase sensitive, but on the figure all the peaks are the same colour? Why is that?
Answer: Thanks for your good suggestion. a) The zoomed aromatic region of HSQC and HMBC spectra have been included after the figure of each HSQC and HMBC spectrum in Supplementary information.b) The original data of HSQC spectrum are in different colors, but it is inconvenient to observe the correlations of C with H. Thus, we processed with MestReNova software to form the cross-section as showed in HSQC spectrum in Supplementary information.
Reviewer 2 Report
It is a very interesting work where they describe very adequately the isolation and characterization of four novel dopamine enantiomer trimers. I have only 4 corrections or suggestions:
1) In lines 78, 80 and 82 they put HMBC corrections and COSY corrections, should be correlations.
2) Lines 106 and 107 compounds 3 and 4 must be in bold type.
3) It would be positive for the work to add a figure with the structures of compounds 5-14 in the manuscript
4) In the Computational Analysis they do not mention how many conformers they finally calculate, if only the most stable conformation or they do some Boltzmann ponderation.
Author Response
Response to Reviewer 2 Comments
1. In lines 78, 80 and 82 they put HMBC corrections and COSY corrections, should be correlations.
Answer: We are so sorry for our spelling mistakes in lines 78, 80 and 82. We have changed “corrections” to “correlations” in our revised manuscript.
2. Lines 106 and 107 compounds 3 and 4 must be in bold type.
Answer: We are so sorry for our neglect that the font format has some errors. These mistakes have been corrected carefully.
3. It would be positive for the work to add a figure with the structures of compounds 5-14 in the manuscript.
Answer: We have added the structures of compounds 5-14 in Figure 1 in our revised manuscript according to your suggestion.
- In the Computational Analysis they do not mention how many conformers they finally calculate, if only the most stable conformation or they do some Boltzmann ponderation.
Answer: We performed all conformers search, not only the most stable conformation. The conformers search results were complemented in our revised Supplementary information.